# Radiomics Analysis in Characterization of Salivary Gland Tumors on MRI: A Systematic Review

**DOI:** 10.3390/cancers15204918

**Published:** 2023-10-10

**Authors:** Kaijing Mao, Lun M. Wong, Rongli Zhang, Tiffany Y. So, Zhiyi Shan, Kuo Feng Hung, Qi Yong H. Ai

**Affiliations:** 1Department of Health Technology and Informatics, The Hong Kong Polytechnic University, Hung Hom, Kowloon, Hong Kong SAR, China; 2Department of Imaging and Interventional Radiology, The Chinese University of Hong Kong, Shatin, New Territories, Hong Kong SAR, China; 3Paediatric Dentistry & Orthodontics, Faculty of Dentistry, The University of Hong Kong, Hong Kong SAR, China; 4Applied Oral Sciences & Community Dental Care, Faculty of Dentistry, The University of Hong Kong, Hong Kong SAR, China

**Keywords:** radiomics, texture analysis, salivary glands tumor, magnetic resonance imaging, systematic review

## Abstract

**Simple Summary:**

This review systematically evaluated radiomics analysis procedures for characterizing salivary gland tumors (SGTs) on magnetic resonance imaging (MRI). Radiomics analysis showed potential for characterizing SGTs on MRI, but its clinical application is limited due to complex procedures and a lack of standardized methods. This review summarized radiomics analysis procedures, focusing on reported methodologies and performances, and proposed potential standards for the procedures for radiomics analysis, which may benefit further developments of radiomics analysis in characterizing SGTs on MRI.

**Abstract:**

Radiomics analysis can potentially characterize salivary gland tumors (SGTs) on magnetic resonance imaging (MRI). The procedures for radiomics analysis were various, and no consistent performances were reported. This review evaluated the methodologies and performances of studies using radiomics analysis to characterize SGTs on MRI. We systematically reviewed studies published until July 2023, which employed radiomics analysis to characterize SGTs on MRI. In total, 14 of 98 studies were eligible. Each study examined 23–334 benign and 8–56 malignant SGTs. Least absolute shrinkage and selection operator (LASSO) was the most common feature selection method (in eight studies). Eleven studies confirmed the stability of selected features using cross-validation or bootstrap. Nine classifiers were used to build models that achieved area under the curves (AUCs) of 0.74 to 1.00 for characterizing benign and malignant SGTs and 0.80 to 0.96 for characterizing pleomorphic adenomas and Warthin’s tumors. Performances were validated using cross-validation, internal, and external datasets in four, six, and two studies, respectively. No single feature consistently appeared in the final models across the studies. No standardized procedure was used for radiomics analysis in characterizing SGTs on MRIs, and various models were proposed. The need for a standard procedure for radiomics analysis is emphasized.

## 1. Introduction

Salivary gland tumors (SGTs) constitute approximately 2–6.5% of all head and neck tumors, with about 80% originating from the parotid gland [1,2,3]. The most prevalent parotid tumors are benign salivary gland tumors (BSGTs), primarily pleomorphic adenoma (PA) and Warthin’s tumor (WT) [4]. The remaining tumors are malignant salivary gland tumors (MSGTs) [4]. The treatment decisions and prognosis for parotid tumors vary depending on their diverse subtypes [5]. Therefore, accurate pre-operative diagnosis is crucial for clinical decision-making in patients with SGTs. Ultrasound-guided fine-needle aspiration cytological (US-guided FNAC) examination is most commonly used for identifying the nature of SGTs [6]. However, this examination is invasive, and due to the heterogeneity of the SGTs, the accuracy of FNAC in the characterization of SGTs only ranges from 86–95%, with 5–14% remaining unknown, leading to repeating FNAC [6,7,8].

Magnetic resonance imaging (MRI) is widely employed in mapping the SGTs for treatment plans because it provides detailed information on soft tissue. MRI has also demonstrated comparable efficacy to FNAC in characterizing SGTs [9,10,11]. More recently, the introduction of radiomics analysis to medical imaging has brought new approaches for quantitative imaging analysis [12], so it is not surprising that researchers have investigated the performance of radiomics analysis in characterizing SGTs on MRI [13,14,15,16,17,18,19,20,21,22,23,24,25,26,27]. Although these studies showed that radiomics analysis offers great potential for characterizing SGTs on MRI, variations in the proposed radiomics models limit its application in clinical practice. A significant challenge hindering further development in this field is the complexity of the radiomics analysis procedures. Subtle differences in each step may lead to variant results. However, a consensus regarding the standardized procedures for radiomics analysis in this context has not yet been reached.

This systematic review aims to evaluate studies that have assessed the performance of radiomics analysis in characterizing benign and malignant SGTs or PA and WT on MRI. The primary focus is analyzing the methodologies and performances reported in each eligible study to provide a comprehensive summary of the approaches employed during the radiomics analysis procedure. The aim is to contribute to the standardization of a radiomics analysis procedure for characterizing SGTs on MRI, which may facilitate its translation into clinical practice.

## 2. Materials and Methods

### 2.1. Research Strategy

The analysis and inclusion criteria methods were pre-defined and documented for this systematic review. The review protocol was registered at PROSPERO International Prospective Register of Systematic Reviews (ID: CRD 42023446728). This review followed the Preferred Reporting Items for Systematic Reviews and Meta-Analysis (PRISMA) 2020 statement [28].

A systematic literature search was conducted through PubMed, Embase, Web of Science, Scopus, and Cochrane Library. The search encompassed studies published from the inception of the electronic databases up to 20 July 2023. The search terms used were “(salivary gland tumor OR parotid gland tumor) AND radiomics AND (MRI OR magnetic resonance imaging)”. These terms were chosen to ensure the inclusion of all salivary gland tumors and to summarize the methodologies and diagnostic performance of radiomics analysis in characterizing SGTs on MRI. The search results were stored in an Excel spreadsheet (Microsoft 365, Microsoft, New York, NY, USA). Duplicate titles were screened and removed. As the data for this review were obtained solely from previously published studies, Institutional Review Board approval or written patient consent was not required and, therefore, waived.

### 2.2. Literature Selection Criteria

The inclusion criteria were as follows:Original articles published in English.Participants: studies involving patients with SGTs who underwent pre-treatment head and neck MRI scans, including at least T1-weighted (T1W), T2-weighted (T2W), contrast-enhanced T1W (CE-T1W), CE-T2W, or diffusion-weighted imaging (DWI).Comparison: studies reporting the performance of radiomics analysis in characterizing SGTs on MRI.Outcomes: the primary outcome was the performance of radiomics analysis in characterizing benign and malignant SGTs on MRI; the second outcome was the performance of radiomics analysis in characterizing PA and WT on MRI.

The exclusion criteria were as follows:Articles in the form of reviews, guidelines, conference proceedings, or case reports/series.Studies that did not report the area under the curve (AUC) of the radiomics models in characterizing SGTs.Studies with patient populations overlapped with previous studies conducted in the same investigated institution for assessing the same outcomes. The exclusion criteria were based on the publication time, with later studies being excluded.

Two observers (QYHA and KFH) independently screened the records based on the title and abstract, and full-text evaluation was performed for selected records. Any disagreements were resolved through discussion or consultation with a third observer (TYS).

### 2.3. Data Extraction

One observer (KM) extracted the following data from the included studies:Study characteristics: first author, journal name, year of publication, city, patient recruitment period, and study design (prospective or retrospective).Patient characteristics: number of patients in the training, testing, and external datasets, methods for diagnosis of the nature of salivary gland tumors.MRI characteristics: MRI sequences used for analysis.Radiomics analysis procedure: segmentation method, number of features extracted, feature categories, methods for feature selection, number and categories of the selected features, names of the selected features, classifiers used for model build-up, and final model.Outcomes: model performance in training, testing, and external datasets.

A second observer (QYHA) verified the received data in an Excel spreadsheet.

### 2.4. Assessment of Study Quality

All of the eligible studies underwent assessments of study quality by one observer (KM) using the Quality Assessment of Diagnostic Accuracy Studies-2 (QUADAS-2) tool and The Radiomics Quality Score (RQS). The QUADAS-2 evaluates four domains: patient selection, index test, reference standard, and flow and timing [29]. Each domain will be rated as low risk, high risk, or unclear risk of bias according to pre-defined criteria [29]. The RQS evaluates radiomics studies based on 16 components, with a score range of 0 to 36 [12].

### 2.5. Statistical Analysis

The inter-observer agreements for article selection based on titles and abstracts were calculated using Cohen’s kappa coefficients.

## 3. Results

### 3.1. Literature Selection

Figure 1 illustrates the study selection process for the systematic review. The initial search of electronic databases yielded 98 titles. After excluding 56 duplicate titles, 42 titles remained for screening. The relevance and quality of these 42 records were assessed based on pre-defined criteria by reviewing their titles and abstracts. At this stage, 24 records were excluded, leaving 18 for full-text assessment. The agreement between the two reviewers who conducted the title and abstract screening was high, with kappa coefficients of 0.95 and 1, respectively. Subsequently, the full texts of the remaining 18 records were obtained and thoroughly reviewed. Four additional records were excluded for specific reasons: one study did not report AUC [30]; three studies reported an overlapping cohort [26,31,32]. Finally, 14 studies were included in the analysis.

### 3.2. Characteristics of the Eligible Studies

Table 1 and Appendix A provide detailed characteristics of the 14 eligible studies. All studies were retrospective evaluations conducted by 12 institutions (8 located in China, 3 in Italy, and 1 in Iran). They were published from 2020 onward. The number of patients in each study ranged from 31 to 334, in a total of 2122, including 339 who had malignant tumors (286 carcinomas, 19 lymphomas, 6 metastases, 20 other types, and 8 uncategorized classifications) and 1802 had benign tumors (977 PA, 681 WT, 55 adenomas, 66 other types, and 23 uncategorized classifications). The majority of tumors observed in the parotid gland were benign, accounting for 85.57% (1690 out of 1975), while all tumors in the sublingual gland were malignant, representing 100% (8 out of 8) of cases (Table 2). Among the 14 studies, 8 (57.1%) reported that the diagnostic gold standard was based on examinations of surgical specimens or fine-needle aspiration cytology.

### 3.3. Characteristics of the Radiomics Analysis Procedures in the Eligible Studies

Table 3 summarizes the characteristics of the radiomics analysis process in all eligible studies. Overall, the analysis was performed to assess the performances of radiomics analysis in characterizing benign and malignant tumors in 10/14 (71.43%) studies (8 on T1W or T2W images, 4 on CE images, and 3 on DWI) and for PA and WT in 7/14 (50%) studies (6 on T1W or T2W images, 1 on CE images, and 1 on DWI). In total, 5/14 studies (35.71%) reported performing image preprocessing, while only 1 study (7.14%) reported employing data augmentation prior to feature extraction. The region or volume of interest (ROI/VOI) was delineated on the whole tumor in 10 out of 14 studies (71.43%), on 2 tumor slices in 1 study (7.14%), and was not reported in 3 studies. The number of features extracted from the ROIs/VOIs varied across studies, ranging from 29 to 3396.

Inter-observer agreement for the extracted features was assessed in 7/14 studies (50%), with all studies using an intra-class correlation coefficient (ICC) threshold of >0.75 to indicate high repeatability. The most common method for feature selection was the least absolute shrinkage and selection operator (LASSO), used in 8/14 studies (57.14%), followed by analysis of variance (ANOVA) in 3/14 studies (21.43%). Cross-validation or bootstrap techniques were applied in 11/14 studies (78.57%) to enhance the stability of the selected features. Logistic regression was the most commonly employed classifier for building the radiomics model, used in 7/14 studies (50%), followed by support vector machine (SVM) in 6/14 studies (42.86%).

Nine studies reported the final selected features for characterizing benign and malignant SGTs. The number of features selected for building the final model ranged from 3 to 17, a total of 61 features (11 first-order features, 5 shape features, 13 texture features, 28 filter-based features, 2 log-based features, and 2 exponential features). Four studies reported the final selected features for characterizing PA and WT. The number of features selected for building the final model ranged from 4 to 13, a total of 36 features (6 first-order features, 1 shape feature, 7 texture features, and 22 filter-based features). No features included in the final models were found to be present in more than two studies.

Table 4 details the final features used for characterizing benign and malignant SGTs. Table 5 details the characteristics of the selected features from eligible studies.

### 3.4. Performances of Radiomics Analysis in Characterizing SGTs

Thirteen studies reported final radiomics models (Table 1). The performance of these models was assessed through various validation methods, including cross-validation in 4 studies, internal dataset validation in 6 studies, and external dataset validation in 2 studies.

#### 3.4.1. Performances of Radiomics Analysis in Characterizing Benign and Malignant SGTs

For characterizing benign and malignant tumors, the AUCs ranged from 0.74 to 1, of which studies that used only T1W or T2W images achieved AUCs ranging from 0.74 to 0.85, and those that used only DWI images achieved AUCs ranging from 0.76 to 0.89. The highest AUC was achieved using features selected from multi-parametric MRI (DCE-MRI, ADC-map, T2WI) validated on a cross-validation dataset, while the lowest AUC was obtained using T2WI.

Three studies compared the performance of radiomics models built using different classifiers. One study reported that radiomics models constructed with support vector machine (SVM) (AUC 0.893) and logistic regression (LR) (AUC 0.886) outperformed those built with k-nearest neighbors (KNN) (AUC 0.796) [17]. Another study demonstrated that extreme gradient boosting (XGBoost) (AUC 0.857) and SVM (AUC 0.809) outperformed models constructed with decision trees (DT) (AUC 0.730) [18]. In a third study, SVM and Fischer’s linear discriminant analysis (LDA) yielded identical performances (AUC 1.0) [21].

Two studies suggested that models incorporating clinical and radiomics features exhibited superior performance compared to models built solely with clinical or radiomics features [19,20].

Three studies compared the performances of radiomics models constructed using different MRI sequences. It was observed that models utilizing multiple MRI sequences for feature extraction outperformed those utilizing a single sequence [16,21,23]. Additionally, the inclusion of CE sequences did not lead to an improvement in performance [23].

One study demonstrated that a radiomics model constructed using a single feature category outperformed a model utilizing all feature categories [23]. Another study indicated that radiomics models built using features extracted from manual segmentation performed better than those using automatic segmentation [16].
cancers-15-04918-t004_Table 4Table 4The final features in models for characterizing benign and malignant salivary gland tumors.Shape (n = 5)Exponential (n = 2)Logarithm (n = 2)DWI_Original_shape sphericity,DWI_SurfaceArea,DWI_Compactness2,DWI_VoxelValueSum,DWI_Maximum3DDiameterFS_T2WI_exponential_glszm_SmallAreaEmphasis,FS-T2WI_exponential_firstorder_90PercentileT1WI_logarithm_glszm_SmallAreaEmphasis,T1WI_logarithm_ngtdm_Complexity**First Order (n = 11)****Texture (n = 13)****Filter (n = 28)**T2WI_skewness value,T2WI_gray level mean,1% percentile,T1WI_Original_first-order_10 Percentile,95th percentile of Ktrans,Maximum,histogram variance,histogram skewness,95th percentile of WIR,histogram standard deviation,DWI_histogram entropyDWI_Gradient glcm_cluster tendency,DWI_Original glszm_small-area low-gray level emphasis,T2WI_autocorrelation value,DWI_SizeZoneVariability,DWI_LongRun-HighGreyLevelEmphasis angle45 offset7,DWI_RunLengthNonuniformity angle0 offset4,DWI_LongRunHighGreyLevelEmphasis angle90 offset1,CE-T1WI_Original_glszm_HighGrayLevelZoneEmphasis,S(0,1) angular second moment,S(5,5) Entrop,S(1,1,0) Entropy,FS-T2WI_ Original_glcm_Imc2,T2WI_GLRLM featuresDWI_Wavelet-LHL_first-order mean, DWI_Wavelet-LHH_gldm large dependence low-gray level emphasis,DWI_Wavelet-HHL_first-order mean, DWI_Wavelet-HHL_glszm small-area low-gray level emphasis,DWI_Wavelet-LLL_glszm small-area low-gray level emphasis, T2WI_Wavelet-HLH_glrlm_RunEntropy,CE-T1WI_Wavelet-LHL_firstorder_Maximum, T1WI_Wavelet-HLH_glrlm_GrayLevelNonUniformityNormalized,CE-T1WI_Wavelet-LLL_glcm_JointAverage, T1WI_Wavelet-LLL_firstorder_Kurtosis, WavEnHH (s-4),T1WI_Wavelet HLL_glcm_Idn, T1WI_Wavelet LHL_gldm_Dependence entropy,T1WI_Wavelet LHH_gldm_Dependence variance, T1WI_Wavelet LHH_first-order_Energy,T1WI_Wavelet LHH_first-order_Total energy, T1WI_Wavelet HLH_gldm_Small dependence low gray level emphasis,T1WI_Wavelet HLH_glcm_Correlation, T1WI_Wavelet HHL_gldm_Small dependence low gray level emphasis,T1WI_Wavelet HHL_glcm_Correlation, T1WI_Wavelet LLL_first-order_Minimum,FS-T2WI_Wavelet LHL_first-order_Mean, FS-T2WI_Wavelet LHL_ngtdm_Busyness,FS-T2WI_Wavelet-HLH_gldm_Dependence entropy, T2WI_wavelet_HLH_glcm_JointEnergy,FS-T2WI_Wavelet HLH_glszm_Gray level nonuniformity normalized,FS-T2WI_Wavelet LLL_first-order_Kurtosis, T2WI_wavelet_LHL_gldm_LargeDependenceEmphasis,CE: contrast-enhanced, DWI: diffusion-weighted imaging, FS: fat saturation, GLCM: gray-level co-occurrence matrix, GLDM: gray-level dependence matrix, GLRLM: gray-level run length matrix, GLSZM: gray-level size zone matrix, H: high-pass filter, L: low-pass filter, NGTDM: neighboring gray tone difference matrix, T1WI: T1-weighted image, T2WI: T2-weighted image, WIR: wash-in rate.
cancers-15-04918-t005_Table 5Table 5Detailed characteristics of the selected features.


Number of Features (Percentage)MRI sequenceBT vs. MTT1WI15 (24.59%)T2WI7 (11.48%)CE-T1WI3 (4.92%)FS-T2WI8 (13.11%)DWI17 (27.87%)Uncategorized11 (15.49%)PA vs. WTT1WI1 (2.78%)T2WI11 (30.56%)FS-T2WI11 (30.56%)DWI13 (36.11%)


**Number of studies (Percentage)**Feature categoryBT vs. MTTexture features7 (77.78%)Filter-based features5 (55.56%)First-order features4 (44.44%)Shape features2 (22.22%)Logarithm-based features1 (11.11%)Exponential-based features1 (11.11%)PA vs. WTFirst-order features4 (100%)Filter-based features3 (75%)Texture features2 (50%)Shape features1 (25%)BT: benign tumors, CE: contrast-enhanced, DWI: diffusion-weighted imaging, FS: fat saturation, MT: malignant tumors, PA: pleomorphic adenoma, T1WI: T1-weighted image, T2WI: T2-weighted image, WT: Warthin’s tumor.

#### 3.4.2. Performances of Radiomics Analysis in Characterizing PA and WT

For characterizing PA and WT, the AUCs ranged from 0.80 to 0.96, of which studies that used only T1W or T2W images achieved AUCs of 0.80–0.96, and those that used only DWI images achieved an AUC of 0.93. The highest and lowest AUCs were achieved using features selected from T2W images.

Similar to the BT vs. MT studies, three studies indicated that models incorporating clinical and radiomics features outperformed those constructed using only clinical or radiomics features [15,22,25]. One study revealed that radiomics models comprising features extracted from multiple MRI sequences performed better than those utilizing a single sequence [22]. Moreover, one study found no significant difference in the performances of radiomics models constructed using computed tomography (CT) and MRI [24].

### 3.5. Quality Assessment

#### 3.5.1. QUADAS-2

The initial flow of QUADAS-2 assessments is presented in Appendix A. A graphical summary of the QUADAS-2 assessment findings is presented in Figure 2, illustrating that the majority of the included studies had a low risk of bias and high applicability.

#### 3.5.2. RQS

The median score of the RQS was 12.5 out of 36, with a range of 5 to 16 (Appendix A). A total of 3 studies scored below 12, and 7 scored above 12 (Figure 3). In most studies, the imaging protocol was well documented. Multiple segmentations and feature reductions were performed, and discrimination statistics were calculated. Nine studies reported on the current and potential application of the model in a clinical setting. None of the studies conducted phantom studies, segmentations at multiple time points, or prospective validations. Furthermore, no study performed a cost-effectiveness analysis or published the algorithms or dataset.

## 4. Discussion

In this systematic review, we evaluated the procedures for radiomics analysis in characterizing SGTs on MRI based on 14 eligible studies. Ten studies focused on characterizing benign and malignant tumors, reporting AUC values ranging from 0.74 to 1. Seven studies differentiated PA and WT, with AUCs ranging from 0.80 to 0.96. Despite the promising accuracy of radiomics, with most studies achieving AUCs above 0.80, these studies employed various methods for radiomics analysis, and none of the resulting models have been further validated or widely implemented in clinical practice. To note, no features included in the final models were found to be present in more than two studies. This systematic review provides valuable insights into MRI sequence selection, image preprocessing, feature extraction, feature selection, and model development, which can help standardize the procedures for radiomics analysis in characterizing SGTs on MRI.

### 4.1. MRI Sequences Selection

The MRI protocol for SGTs typically includes multiple MRI sequences, such as T1WI, T2WI, and CE images. Functional MRI techniques, such as dynamic contrast-enhanced (DCE) MRI and DWI, have also demonstrated potential for characterizing SGTs and are increasingly included in the MRI protocol for SGTs [33]. In radiomics analysis, it is not surprising that our results showed that models constructed using multiple MRI sequences outperformed those built using a single sequence. However, including radiomics features from functional MRI sequences, such as DCE MRI and DWI, did not significantly improve performance and could potentially increase the computational burden. It is also worth noting that one study indicated that radiomics features derived from CE images did not provide additional value in characterizing SGTs [23]. Other studies in head and neck cancer have demonstrated that non-contrast-enhanced MRIs can be used instead of CE MRIs to develop machine-learning algorithms for clinical tasks [34,35]. Therefore, the value of CE MRI in the context of radiomics characterization needs further evaluation.

### 4.2. Image Preprocessing and Feature Extraction

Image preprocessing and feature extraction play crucial roles in radiomics analysis, but their implementation and reporting were inconsistent across the studies reviewed. Only five studies reported on image preprocessing, while the details of this step were unclear in the remaining studies. Image preprocessing is essential for standardizing images obtained from different institutions with varying protocols. This step is particularly critical but challenging to implement for MRI as MRIs are constructed by weighting the signals corresponding to the magnetic properties of the tissues being imaged. Several common methods have been proposed to address different protocol parameters and improve the quality and consistency of MRI images, including the “μ ± 3σ” method, N4ITK bias field correction, resampling, and Z-score normalization [36,37,38,39]. However, further investigation is needed to determine the effectiveness of each method.

The selection of ROI is an important consideration and depends heavily on the heterogeneity of the tumor as depicted on imaging. Since SGTs can exhibit high heterogeneity, it is suggested to use an ROI covering the whole tumor. Interestingly, one study showed that radiomics models built using features from manual segmentation outperformed those using automatic segmentation for characterizing benign and malignant SGTs on MRI [16]. This may be due to the current limitations of automatic segmentation methods for accurately segmenting SGTs. While manual tumor segmentation can be time-consuming, there is a need for tools that provide accurate and consistent ROI segmentation.

On the other hand, in diseases with low incidences, such as SGTs, the number of features extracted for analysis should be carefully considered. The reviewed studies had total patient numbers ranging from 31 to 334, and the total number of extracted features ranged from 29 to 3396. Most studies extracted over 400 features. These increased the risk of overfitting the model. One study notably demonstrated that the radiomics model built up by the selection from over 1000 radiomics features did not outperform that from 91 features [23]. Furthermore, this study showed that feature category should also be considered when radiomics analysis was performed in characterizing SGTs on MRI [23]. It is also important to evaluate the repeatability of the extracted features, which can be significantly influenced by inter-observer variability in ROI delineation. Inter-observer agreement for the features should be assessed before further analysis. Among the reviewed studies, seven evaluated feature repeatability and reported a high inter-class correlation coefficient (ICC) above 0.75 to confirm the repeatability of the features.

### 4.3. Feature Selection and Model Build-Up

The review showed the LASSO method as the most commonly used feature selection method in the analyzed studies, with 8/14 studies employing it. LASSO is a regularization-based method that can effectively remove irrelevant or redundant features by shrinking their coefficients to zero [40]. This helps reduce the risk of overfitting and improves the model’s generalizability [40]. However, it is crucial to consider the potential impact of correlation among features and the choice of regularization parameter on the performance of LASSO. Other feature selection methods, such as minimum redundancy maximum relevance (mRMR), were also utilized in some studies [41]. mRMR shares similar strengths with LASSO but may be sensitive to feature settings or the presence of outliers [42]. Some studies combined multiple feature selection methods to exploit their complementary strengths [18,43,44]. However, the stability and reliability of the selected features should be carefully evaluated. In addition to cross-validation and bootstrapping used in 11 studies, the bagged-boosted repeated elastic net technique (BB-RENT) can effectively select features with high stability and reliability [45]. However, the performance of models built up using features selected by these methods still needs further evaluation [45].

LR and SVM were the most commonly used classifiers for building radiomics models to characterize SGTs on MRI. However, the best-performing classifier varied among the studies. In three studies that compared multiple classifiers, SVM and LR outperformed KNN in one study, XGBoost and SVM performed better than DT in another study, and SVM and LDA performed similarly in the third study [17,18,21]. Model performance was validated using cross-validation in four studies, the internal dataset in six studies, and the external dataset in two studies. The AUCs of models validated by cross-validation were generally higher than those using the internal dataset but lower than those using the external dataset. Seven studies included texture features in the final model, indicating they may be the most likely useful category in characterizing benign and malignant SGTs. However, no single feature was consistently included in the final models across multiple studies, emphasizing the need for the improved generalizability of radiomics models for clinical applications.

### 4.4. Suggestions

Based on the findings of this review, the following suggestions can be made to facilitate radiomics analysis and the development of radiomics models in characterizing SGTs on MRI:Focus on non-contrast-enhanced MRIs.Implement image preprocessing.Limit the number of features extracted and consider the feature categories.Evaluate inter-observer agreement for the extracted features and select those with high repeatability for further analysis.Use multiple feature selection methods.Ensure feature stability by different approaches.Build models using different approaches and identify the best model.Validate models using at least cross-validation dataset with/without internal or external datasets.Report the final models for future validations.Open datasets (not necessarily the original images) for future validations.

Implementing these recommendations will enhance the standardization, reproducibility, and generalizability of radiomics analysis and radiomics models in characterizing SGTs on MRI, ultimately facilitating their broader use in clinical practice.

### 4.5. Limitations

This review has limitations that may result in inherent heterogeneity and publication bias. Firstly, all studies included in this systematic review were retrospective studies. Due to the retrospective nature, the concerns regarding the risk of bias in the included studies could not be avoided. However, the results from QUADAS-2 showed that most of the studies had a low risk of bias and high applicability. Secondly, a meta-analysis that evaluated the performance of radiomics models in characterizing SGTs on MRI was inappropriate due to the significant heterogeneity of the radiomics models among the eligible studies. Thirdly, according to the registered study protocol, eligible studies that were published after July were not included in the analysis. Furthermore, the added value of radiomics analysis to clinical practice remains underreported, and no studies have analyzed cost-effectiveness. Therefore, it is recommended that further studies be conducted to evaluate radiomics analysis using standardized procedures for characterizing SGTs on MRIs.

## 5. Conclusions

Previous studies have demonstrated the potential of radiomics analysis in characterizing SGTs on MRI. However, the lack of standardized procedures for implementing radiomics analysis across these studies may have led to the limited generalizability of the final radiomics models, thereby restricting their application in clinical practice. To develop radiomics models that can be widely utilized for characterizing SGTs on MRI in the future, it is crucial to establish a consensus on the procedures for conducting radiomics analysis.

## Figures and Tables

**Figure 1 cancers-15-04918-f001:**
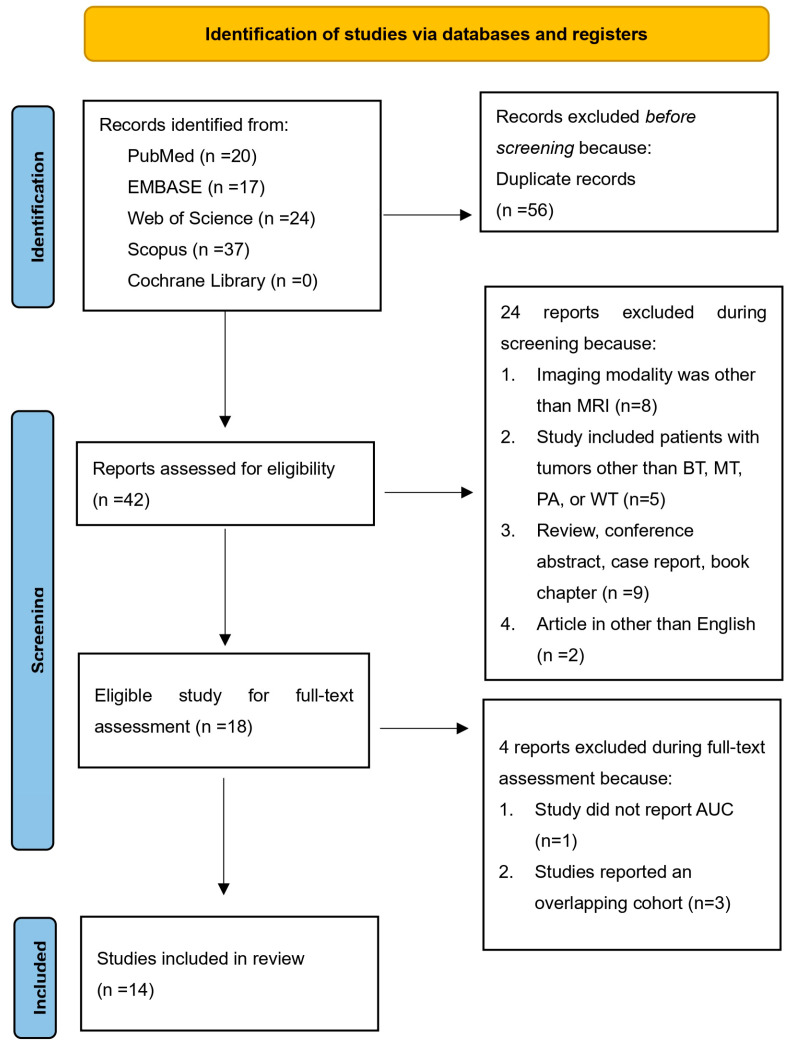
Flowchart of study selection.

**Figure 2 cancers-15-04918-f002:**
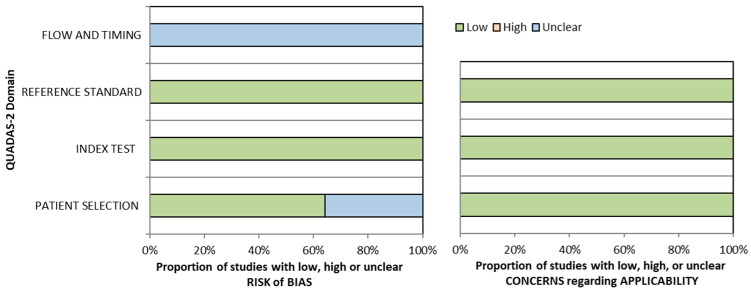
Graphical summary of QUADAS-2.

**Figure 3 cancers-15-04918-f003:**
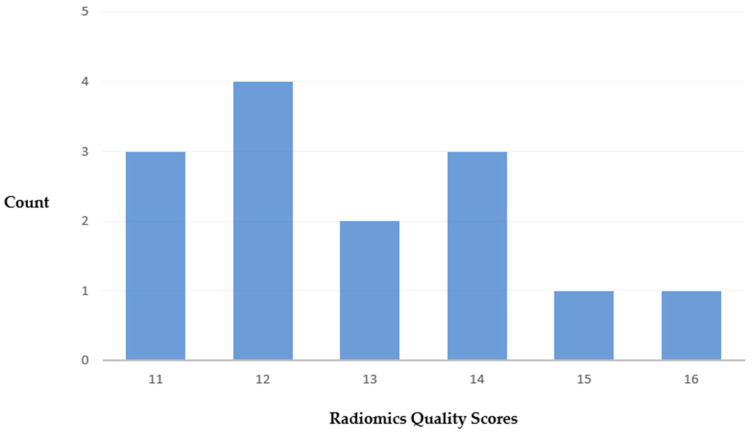
Distribution of total RQS.

**Table 1 cancers-15-04918-t001:** The characteristics of each included study.

Study ID	Year	Number of Cases (n)	Number of External Cases (n)	Number of Features for Different Tasks (Selected/Extracted)	Final Models	BT vs. MT (AUC)	PA vs. WT (AUC)
1 [13]	2022	130	NA	BT vs. MT 8/944 PA vs. WT 13/944	BT vs. MT: the LDA model based on 8 features on DWI, PA vs. WT: 13 features on DWI	0.7637	0.925
2 [14]	2020	75	NA	BT vs. MT 5/29 PA vs. WT 4/29	BT vs. MT: SVM with a radiomics signature with 5 features on T2WI PA vs. WT: SVM with 4 features on T2WI	0.7365	0.8179
3 [15]	2021	127	52	PA vs. WT 12/1702	The radiomics nomogram incorporating the age and radiomics signature with 12 radiomics features on T1WI and FS-T2WI	NA	0.953
4 [16]	2021	57	NA	BT vs. MT PA vs. WT NA/289	BT vs. MT: radiomics models based on texture analysis through manual segmentation on T1-T2WI; PA vs. WT: radiomics models based on texture analysis through manual segmentation on T2WI	0.927	0.802
5 [17]	2020	269	NA	BT vs. MT 8/396	Eight features with LR or SVM models on DWI	0.893	NA
6 [18]	2022	298	NA	BT vs. MT 6/3396	Six features with XGBoost on the combination of T2WI, T2WI, and CE-T1WI	0.857	NA
7 [19]	2021	109	NA	BT vs. MT 5/1059	Model with clinical data + 2D and 3D biomarkers (5 features) on T1-T2WI	0.85	NA
8 [20]	2021	115	35	BT vs. MT 17/1702	Radiomics nomogram incorporating the clinical factors and radiomics signature (17 features from T1WI and FS-T2WI)	0.952	NA
9 [21]	2022	31	NA	BT vs. MT 8/77	Radiomics analysis of the combination of T2WI, ADC-map, and DCE-MRI parametric maps with SVM or LDA with 8 features	1	NA
10 [22]	2021	252	NA	PA vs. WT 7/429 T1WI 8/414 T2WI 8 T1-2WI	T1-2WI radiomics model using MLR with selected features	NA	0.952
11 [23]	2022	91	NA	BT vs. MT 4/1015	A combination of T1WI + logarithm and FS-T2WI + exponential features with LR classifier	0.846	NA
12 [24]	2021	334	NA	PA vs. WT NA/30	NA	NA	0.911
13 [25]	2022	117	NA	PA vs. WT 8/971	The radiomics–clinical model with 8 features on T2WI	NA	0.962
14 [27]	2023	117	NA	BT vs. MT 2/851	SVM with 2 radiomics features on T2WI and 4 inflammatory biomarkers	0.79	NA

ADC: apparent diffusion coefficient, AUC: area under curve, BT: benign tumors, CE: contrast-enhanced, DCE: dynamic contrast-enhanced, DWI: diffusion-weighted imaging, FS: fat saturation, LDA: linear discriminant analysis, LR: logistic regression, NA: not available, MT: malignant tumors, PA: pleomorphic adenoma, SVM: support vector machine, T1WI: T1-weighted image, T2WI: T2-weighted image, WT: Warthin’s tumor.

**Table 2 cancers-15-04918-t002:** Distribution of malignant and benign tumors across different salivary glands.

Salivary Gland	MT	BT
Parotid	285	1690
Submandibular	14	21
Sublingual	8	-
Minor	4	1
Uncategorized	28	89

BT: benign tumors, MT: malignant tumors.

**Table 3 cancers-15-04918-t003:** Overview summary of the included studies.

Characteristics			Number of Studies (Percentage)
MRI sequence	BT vs. MT(10)	T1WI or T2WI	8 (80% for BT vs. MT)
		CE	4 (40% for BT vs. MT)
		FS	2 (20% for BT vs. MT)
		DCE	1 (10% for BT vs. MT)
		DWI	3 (30% for BT vs. MT)
	PA vs. WT(7)	T1WI or T2WI	6 (85.71% for PA vs. WT)
		CE	1 (14.29% for PA vs. WT)
		FS	1 (14.29% for PA vs. WT)
		DWI	1 (14.29% for PA vs. WT)
Segmentation	Manual segmentation		12 (85.71%)
	Semi or automatic segmentation		3 (21.43%)
Region/volume of interest	Two slices		1 (7.14%)
	Whole tumor		10 (71.43%)
	Not reported		3 (21.43%)
Image preprocessing	Reported		5 (35.71%)
Data augmentation	Reported		1 (7.14%)
Inter-observer agreement for feature selection	Reported		7 (50%)
Validation for feature selection	Reported		11 (78.57%)

BT: benign tumors, CE: contrast-enhanced, DCE: dynamic contrast-enhanced, DWI: diffusion-weighted imaging, FS: fat saturation, MT: malignant tumors, PA: pleomorphic adenoma, T1WI: T1-weighted image, T2WI: T2-weighted image, WT: Warthin’s tumor.

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
