# Peer review of "Radiomics Analysis in Characterization of Salivary Gland Tumors on MRI: A Systematic Review"

_cancers, 2023, doi:10.3390/cancers15204918_

Round 1

Reviewer 1 Report

The paper is well written, innovative and especially it is full of detailed informations. The supplementary material is also adequate. A comprehensive literature review is performed. Overall it has good scientific relevance. And most important it can be a valid support to readers who are less expert in the  field of radiomic analysis for SGT.For these reasons I recommend the publication in present form

Author Response

Response: Thank you for your positive feedback and recommendation for publication.

Reviewer 2 Report

The authors systematically evaluated previous reports about radiomics on MRI for salivary gland tumors (SGTs). Through reviewing 14 papers proper for the criteria, they demonstrated that they had low risk of bias and high applicability, but no consistency in methodology of MRI-based radiomics research for SGTs. Their review suggests the future load map for MRI-based radiomics of SGTs toward practical usage.

In Figure 2, the right panel, all bars are green (low applicability). ?

In the 2nd line of paragraph 3.5.2. RQS, Figure 1 should be Figure 3.

Author Response

Response: Thank you for your feedback and positive evaluation of our study.

In Figure 2, the right panel, all bars are green (low applicability). ?

Response: The green bars in Figure 2, located in the right corner, illustrate a low level of concerns regarding the applicability. The figure states, "Proportion of studies with low, high, or unclear CONCERNS regarding applicability."

In the 2nd line of paragraph 3.5.2. RQS, Figure 1 should be Figure 3.

Response: Thanks for your kind reminder. The change has been made.

Reviewer 3 Report

The present article systematically evaluated radiomic analysis procedures for charac- 15 terizing salivary gland tumors (SGTs) on magnetic resonance imaging (MRI). Radiomic analysis showed in the present study the potential for characterizing SGTs on MRI, but its clinical application is limited due to com- 17 plex procedures and a lack of standardized methods. A number of  radiomic procedures, focusing on reported methodologies and performances, and proposed potential standards for the procedures of radiomics analysis, were presented.

I appreciate that the authors registered the research in Prospero.

How were the duplicates screened?

Please add in the discussion chapter more recent published articles.

Moderate revision.

Author Response

Response: Thank you for your valuable comments on our manuscript.

How were the duplicates screened?

Response: All study titles are sorted alphabetically in Excel and screened for duplicates.

Please add in the discussion chapter more recent published articles.

Response: This study systematically reviewed papers published online before July 2023. The research proposal was prospectively designed and registered in "PROSPERO". It would not follow the proposal if we added the recently published papers to the manuscript. However, we would like to revise the discussion to point out this.

Reviewer 4 Report

This is a systematic review about radiomics analysis in the characterization of salivary gland tumors (SGT) on MRI. A systematic literature search was conducted through PubMed, Embase, Web of Science, Scopus, and Cochrane Library. The authors found that no standardized procedure was used for radiomics analysis in characterizing SGTs on MRI, and various models were proposed.

Materials and methods are adequately described.

The discussion is appropriate and limitations are reported. References are adequate.

The paper is well written. However, some issues remain.

All the acronyms must be explained at their first appearance in the text and/or the abstract.

Author Response

Response: Thank you for your valuable feedback on our manuscript.

All the acronyms must be explained at their first appearance in the text and/or the abstract.

Response: Thanks for your kind reminder. The change has been made.